# Cross-Sectional Associations of Self-Reported Social/Emotional Support and Life Satisfaction with Smoking and Vaping Status in Adults

**DOI:** 10.3390/ijerph191710722

**Published:** 2022-08-28

**Authors:** Zidian Xie, Francisco Cartujano-Barrera, Paula Cupertino, Dongmei Li

**Affiliations:** 1Department of Clinical and Translational Research, University of Rochester Medical Center, Rochester, NY 14642, USA; 2Department of Public Health Sciences and Cancer Center, University of Rochester Medical Center, Rochester, NY 14642, USA

**Keywords:** smoking, vaping, social/emotional support, life satisfaction, smoking cessation

## Abstract

This study aimed to examine the cross-sectional association of self-reported social/emotional support and life satisfaction with smoking/vaping status in US adults. The study included 47,163 adult participants who self-reported social/emotional support, life satisfaction, and smoking/vaping status in the 2016 and 2017 BRFSS national survey data. We used multivariable weighted logistic regression models to measure the cross-sectional association of self-reported social/emotional support and life satisfaction with smoking/vaping status. Compared to never users, dual users and exclusive smokers were more likely to have low life satisfaction, with an adjusted odds ratio (aOR) = 1.770 (95% confidence interval [CI]: 1.135, 2.760) and an aOR = 1.452 (95% CI: 1.121, 1.880) respectively, especially for the age group 18–34. Exclusive cigarette smokers were more likely to have low life satisfaction compared to ex-smokers (aOR = 1.416, 95% CI: 1.095, 1.831). Exclusive cigarette smokers were more likely to have low social/emotional support (aOR = 1.193, 95% CI: 1.030, 1.381) than never users, especially those aged 65 and above. In addition, exclusive cigarette smokers were more likely to have low social/emotional support than ex-smokers, with an aOR = 1.279 (95% CI: 1.097, 1.492), which is more pronounced among the age group 18–34, as well as 65 and above. Our results suggest that life satisfaction and social/emotional support may play important roles in smoking and vaping, which should be incorporated into behavioral interventions to reduce tobacco use.

## 1. Introduction

Tobacco smoking kills more than 8 million people each year worldwide [1]. In 2019, approximately 34.1 million adults were cigarette smokers in the United States (US) [2]. While there has been a decline in smoking prevalence in recent years, e-cigarette use (vaping) has become increasingly popular, especially among youths (ages 12 to 18) and young adults [3,4]. In 2019, 4.5% of adults in the US were current e-cigarette users [2]. While the long-term health effects of vaping remain to be determined, vaping is associated with many health effects, including respiratory and cardiovascular diseases [5,6,7,8,9,10]. Therefore, it is of utmost importance to understand psychosocial constructs related to smoking and vaping to inform future smoking and vaping cessation interventions.

Smoking and vaping are related to poor mental health, including quality of life, depression, and anxiety [11,12,13,14]. Nicotine triggers the release of dopamine, which plays a significant role in emotional responses and reward-motivated behavior [15]. Therefore, tobacco use by smoking or vaping might be related to the relaxation reduction of stress and anxiety. On the other hand, smoking and vaping have both been associated with psychiatric symptoms (such as depression) [16,17,18,19]. Defined as an individual’s cognitive evaluation of their life, life satisfaction has strong interactive effects on mental health, as one of the main dimensions [20,21]. Several previous studies have shown that low life satisfaction is highly associated with substance use, including marijuana, alcohol, cannabis, and smoking [22,23], as well as other addictive behaviors [24].

While most smokers are willing to quit (55.1% of adult smokers in 2018), the success rates are very low (7.5% in 2018), even with different treatments, including nicotine replacement therapy [3]. Once addicted to nicotine, it is hard to stop, due to the negative emotional responses involved in the withdrawal effect of nicotine [25]. Social and emotional support is defined as the social resources that one receives as a result of empathetic, caring, and reassuring communication from non-professionals [26]. As moderators of stress, social and emotional support are usually regarded as an essential factor for smoking cessation, although their exact role remains controversial [27,28]. Several studies have shown that positive social and emotional support are associated with successful smoking cessation and prevention of relapse [29,30,31]. There is one study that showed that adult smokers in South Korean rural areas had high levels of social support [32]. However, whether life satisfaction and social/emotional support might be associated with vaping remains undetermined.

This study aimed to examine the cross-sectional association of self-reported life satisfaction and social/emotional support with smoking and vaping status, using the US nationally represented survey data. Because vaping is more prevalent among young adults [33], the present study will also expand the understanding of the association of smoking and vaping with social/emotional support and life satisfaction across age groups. Results from the present study are intended to inform the development of future interventions designed to promote both smoking and vaping prevention and/or cessation among US adults.

## 2. Materials and Methods

### 2.1. Study Participants

The Behavioral Risk Factor Surveillance System (BRFSS) is an annual interview survey, conducted through landlines and cell phones by the Centers for Disease Control and Prevention (CDC) in the US, which collects the information about basic demographics (such as age, gender, education, and income), health-related risk behaviors, and chronic health conditions among adult participants in all 50 US states, as well as the District of Columbia and the three US territories (including Puerto Rico, Guam, and U.S. Virgin Islands) [34]. The BRFSS survey was designed in a way such that data from different years could be easily combined to study the association of e-cigarette use with respiratory diseases (such as COPD and asthma) and other health conditions [10,35,36,37,38,39]. Since the survey data in 2016 and 2017 contain the most recent and similar interview questions related to smoking and vaping, especially life satisfaction and emotional support, we decided to combine the 2016 (486,303 participants) and 2017 (450,016 participants) BRFSS data in our analysis. The 2016 and 2017 BRFSS data are publicly available from the CDC website: https://www.cdc.gov/brfss/annual_data/annual_data.htm (accessed on 16 February 2021).

### 2.2. Vaping and Smoking Categories

Based on their current smoking and vaping status, as well as past smoking experience, we grouped the adult participants into six smoking and vaping categories: (1) exclusive smokers: currently established smokers (have smoked at least 100 cigarettes in their entire life and now smoke every day or some days) who were not currently established vapers; (2) ex-smokers: previous smokers (have smoked at least 100 cigarettes in their entire life and now do not smoke cigarettes at all) who were not currently established vapers; (3) current vapers who were ex-smokers: currently established vapers (using e-cigarettes every day or some days) who were ex-smokers; (4) exclusive vapers: currently established vapers who were never smokers; (5) dual users: currently established smokers and currently established vapers; (6) never users: never smokers (have smoked less than 100 cigarettes in their entire life, and now do not smoke cigarettes at all) who also were not currently established vapers.

### 2.3. Main Measures

The primary outcome variables of interest in this study included self-reported life satisfaction and social/emotional support. The variable for life satisfaction came from the survey question, “In general, how satisfied are you with your life?”. We classified it into two levels, one as “low life satisfaction”, with the response of “dissatisfied” or “very dissatisfied”, and the other as “high life satisfaction”, with the response of “satisfied” or “very satisfied.” The variable for social/emotional support came from the survey question, “How often do you get the social and emotional support you need?”. We classified it into two levels: one as “high emotional support”, with the answer of “always” or “usually,” and the other as “low emotional support”, with the response of “sometimes” and “rarely”.

### 2.4. Covariates

Using the purposeful variable selection method [10,40], covariates that significantly contributed to the models were selected and controlled for our statistical models, including age, gender, employment status, self-reported general health categories, and self-reported mental health (including stress, depression, and problems with emotions). To examine if there is an age effect on the association of smoking and vaping with life satisfaction and social/emotional support, with the consideration of the sample size, we divided adult participants into three age groups, including “18–34,” “35–64,” and “65+.” It has been shown that compared to adults within other age categories, adults in the 18–34 category have higher odds of e-cigarette use, as well as different socioeconomic statuses, for example, poverty and marital status, which might affect their life satisfaction [33,41,42]. Except for mental health, all other covariates are categorical variables.

### 2.5. Statistical Analysis

Weighted frequency distributions were calculated to determine the association of smoking and vaping status with each covariate. After adjusting for those selected covariates, multivariate weighted logistic regression models were used to assess the association of tobacco and vaping status with the outcome variables (life satisfaction and social/emotional support). To account for the complex sampling design in the BRFSS survey study, the variable _LLCPWT as the final weight, the stratification variable _STSTR, and the clustering variable _PSU for each adult participant were included in the statistical models. The final weight was half of the weight from each year (2016 and 2017) to ensure the weighted sample size was equivalent to the population size [10]. To measure the association of smoking and vaping status with life satisfaction and social/emotional support, multivariable weighted logistic regression models with PROC SURVEY procedures in SAS V9.4 (SAS Institute Inc., Cary, NC, USA) were used to calculate the adjusted odds ratios (aORs) and 95% confidence intervals (CIs). The Taylor series linearization method was used to calculate the standard deviations. All significance tests were two-sided, with a significance level of 5%.

## 3. Results

### 3.1. Demographic Characteristics of Survey Participants

From the combined 2016 and 2017 BRFSS survey data, there were 47,163 out of 936,319 adult participants included in our study who indicated their smoking and vaping status, as well as provided valid responses to our main measures about life satisfaction and social/emotional support. Among them, 6121 (12.98%) were exclusive smokers, 806 (1.71%) were dual users, 13,376 (28.36%) were ex-smokers, 26,198 (55.55%) were never users, 454 (0.96%) were current vapers who were ex-smokers, and 208 (0.44%) were exclusive vapers.

As shown in Table 1, the majority of participants aged between 35 and 64 years in all smoking/vaping categories except exclusive vapers. The majority of exclusive vapers were young, aged 18–34 (90.38%) and never married (76.75%). While there was a similar proportion of males and females in other categories (such as dual users, exclusive smokers, ex-smokers, and never users), there were more males than females in current vapers who were ex-smokers (60.87% vs. 39.13%) or exclusive vapers (67.87% vs. 32.14%). Among exclusive vapers, 28.80% were students, which was higher than in other smoking/vaping categories. Of ex-smokers, 30.32% were retired, higher than in other smoking/vaping categories.

By comparison, never users and ex-smokers had the best overall self-reported mental health, with 2.49 days and 2.76 days (out of 30 days) of having bad mental health. Relatively, exclusive vapers and current vapers who were ex-smokers had worse self-reported mental health, with 5.45 and 5.16 days of having bad mental health. Exclusive smokers and dual users had the worst self-reported mental health, with 6.09 and 8.06 days of having bad mental health.

### 3.2. Association of Life Satisfaction with Smoking/Vaping Status

Exclusive smokers and dual users were more likely to receive lower social/emotional support and life satisfaction than other groups (Table 1). For example, 15.88% of dual users and 12.18% of exclusive smokers had low life satisfaction, while it was only 4.43% for ex-smokers and 3.40% for never users. From the weighted multivariate logistic regression models, the adjusted odds ratios of the other five smoking and vaping categories relative to never users for different covariates were calculated (Appendix A). As shown in Table 2, compared to never users, while other smoking/vaping categories did not show any significant difference, dual users and exclusive smokers had significantly higher odds ratios of low life satisfaction with an aOR = 1.770 (95% CI: 1.135, 2.760) and an aOR = 1.452 (95% CI: 1.254, 5.896). In addition, exclusive smokers had a higher odds ratio of low life satisfaction than ex-smokers (aOR = 1.416, 95% CI: 1.095, 1.831).

As shown in Table 2, similar to all adults, in the age group 18–34, dual users and exclusive smokers had higher odds ratios of low life satisfaction than never users, with an aOR = 2.719 (95% CI: 1.254, 5.896) and an aOR = 2.457 (95% CI: 1.439, 4.193), respectively. Compared to never users, in age group 65+, exclusive vapers and current vapers who were ex-smokers had significantly lower odds ratios of low life satisfaction, with an aOR = 0.013 (95% CI: 0.001, 0.161) and an aOR = 0.104 (95% CI: 0.014, 0.786). However, the sample size for these categories was very small. For example, there were 62 subjects for current vapers who were ex-smokers and in the age group 65+, and only two subjects for exclusive vapers in the age group 65+ (Appendix A).

### 3.3. Association of Social/Emotional Support with Smoking/Vaping Status

To understand if social and emotional support might be associated with smoking/vaping status, we calculated the odds ratios of having low social/emotional support between different smoking/vaping categories. As shown in Table 3, compared to never users, only exclusive cigarette smokers showed a significantly higher odds ratio of low social/emotional support, with an aOR = 1.193 (95% CI: 1.030, 1.381). Moreover, exclusive smokers showed a significantly higher odds ratio of low social/emotional support than ex-smokers (aOR = 1.279, 95% CI: 1.097, 1.492). In addition, we examined the association of social/emotional support with smoking/vaping status in different age groups. Exclusive smokers in the age group 65+ showed a higher odds ratio of low social/emotional support than never users (aOR = 1.385, 95% CI: 1.061, 1.808). Compared to ex-smokers, exclusive smokers in the age groups 18–34 and 65+ showed a higher odds ratio of low social/emotional support, with an aOR = 1.542 (95% CI: 1.003, 2.372) and an aOR = 1.470 (95% CI: 1.125, 1.920). Compared to never users, exclusive vapers age 65+ (only two subjects) showed a lower odds ratio of low social/emotional support (aOR = 0.014, 95% CI: 0.003, 0.071).

## 4. Discussion

This study aimed to examine possible cross-sectional associations between smoking/vaping status and life satisfaction and social/emotional support among US adults. Our results showed that dual users and exclusive smokers were more likely to have low life satisfaction than never users, especially among young adults (age group 18–34). In contrast, vapers in the age group 65+ were more likely to be satisfied with their life, but the sample size was relatively small. Exclusive smokers were more likely to have low life satisfaction than ex-smokers. Compared to never users, exclusive smokers were more likely to have low social/emotional support, especially among the elders (age group 65+). Exclusive smokers were more likely to have lower social/emotional support than ex-smokers among age groups 18–34 and 65+.

In this study, we showed that, compared to never users, smokers and dual users had significantly higher odds of low life satisfaction among young adults but not among old adults. Previous studies showed that low life satisfaction is significantly associated with smoking in adolescents [22,43]. Hoping to improve their life quality, low life satisfaction might motivate individuals (especially young adults) to engage in health risk-related behaviors, such as tobacco use, alcohol, and drug abuse [44]. On the other hand, these health risk behaviors might lead to low life satisfaction [45]. However, our cross-sectional study does not determine the causal direction between low life satisfaction and smoking or vaping. In addition, our results showed that exclusive smokers were more likely to have low life satisfaction than ex-smokers. Several studies showed that ex-smokers were much happier with their lives or had a better quality of life than when they were smokers [46,47], supporting a notion that quitting smoking could improve life satisfaction. Our study showed that vapers (exclusive vapers and current vapers who were ex-smokers) had lower odds of low life satisfaction than never users, but the differences were not statistically significant. A lack of association of vaping with life satisfaction could be due to either the relatively small sample size or the possibility that there is no association between vaping with life satisfaction. The association between vaping and life satisfaction should be further validated in the future.

Our results showed that current smokers received significantly lower social and emotional support than never users, especially among the age group 65+. More importantly, current smokers had lower social and emotional support than ex-smokers. These results might suggest that low social and emotional support might be related to smoking. More social and emotional support might help with smoking cessation, which is consistent with previous findings [29,48,49,50]. A proposed model suggests that social and emotional support might help with the smoking cessation by alleviating stress (also known as stress buffering) [51,52]. When facing daily stress or withdrawal symptoms, with social and emotional support, one does not need to seek substance use for coping with stresses and therefore continue the smoking cessation process [53]. It has been shown that online help is popular for providing social and emotional support for smokers [49,54]. While it was reported that most smokers have attempted to quit smoking on their own [55], social and emotional support might be an effective strategy to help this process, such as social support from their family members, especially their romantic partners [56]. With the development of the internet and electronic technologies, social and emotional support from friends and peers through social media (such as Twitter, Reddit, and Facebook) might be another effective approach for smoking cessation, which awaits further exploration and validation in the future [57]. While several social and emotional support approaches (such as friends, family members, and the community) could help with smoking/vaping cessation, it is important to determine which ones might be more effective in further studies. In our study, we did not observe a significant association of social/emotional support with current vaping status, which could be partially due to the relatively small sample size. Another possible explanation is that unlike smoking, vaping is not associated with social/emotional support. Considering the current vaping epidemic [58], it is important to determine its potential association with social/emotional support in future studies with a large sample size.

There were several limitations to our study. First, some of the smoking and vaping groups in our study had a relatively small sample size. An example would be the exclusive vapers and current vapers who were ex-smokers, especially in the old age group (65+). Therefore, the associations of life satisfaction and social/emotional support with vaping were not conclusive, especially among the youth, which requires further investigation. Second, our study was based on cross-sectional data, which did not determine the causal effect between life satisfaction or social/emotional support and smoking/vaping status. We also could not determine the longitudinal change in life satisfaction when smokers quit smoking. Third, some other important information about smoking and vaping (such as if they ever attempted to quit smoking, if there are current smokers or vapers in their family/friends, the number of cigarettes per day, the number of days used in the past month, and how long they have been smoking or vaping) were not included in our models, due to the unavailability of these data in the BRFSS survey, which might introduce some biases into our final results. In addition, in this study, there was no information on other tobacco product use, such as cigars or the waterpipe in the BRFSS data. Thus, our study only considered smoking and vaping status, which could bias our results. There was a possibility that some participants might have answered both the 2016 and 2017 BRFSS survey questionnaire. However, we were not able to identify those participants due to the anonymous property of the BRFSS survey, which might bring some bias to our analysis results. Finally, in this study we grouped survey participants into three large age groups. However, the level of life satisfaction and social/emotional support depended on the age [59,60]. Even with the same age group (such as age group 18–34), participants with different ages might have different levels of life satisfaction and social/emotional support [59]. Therefore, our age grouping method might introduce some biases.

## 5. Conclusions

In this study, using the US national survey data with 47,163 subjects, we showed that adult smokers are more likely to have low life satisfaction and low social/emotional support compared to never users or ex-smokers, suggesting the potential role of life satisfaction and social/emotional support in the current smoking status. Together, our study provided some preliminary but valuable information on potential smoking maintenance, which could be sources for interventions for smoking or even vaping cessation.

## Figures and Tables

**Table 1 ijerph-19-10722-t001:** The distribution of co-variates within smoking/vaping status.

		Current Vaping and Smoking Status (% with 95% CI)						
		Exclusive Smokers	Dual Users	Current Vapers Who Were Ex-Smokers	Exclusive Vapers	Ex-Smokers	Never Users	*p*
Variables	Levels	(*n* = 6121)	(*n* = 806)	(*n* = 454)	(*n* = 208)	(*n* = 13,376)	(*n* = 26,198)	
Age								<0.0001
	18–34	28.56 (27.12, 30.07)	44.52 (40.91, 48.45)	35.44 (31.64, 39.70)	90.38 (89.26, 91.52)	13.05 (12.08, 14.11)	31.63 (30.86, 32.42)	
	35–64	58.22 (57.53, 58.93)	50.46 (49.00, 51.97)	57.85 (54.90, 60.96)	9.17 (5.98, 14.03)	51.73 (51.08, 52.39)	49.60 (49.09, 50.11)	
	65+	13.22 (12.59, 13.88)	5.01 (4.31, 5.83)	6.71 (5.59, 8.05)	0.45 (0.13 (1.66)	35.22 (34.80, 35.65)	18.77 (18.42, 19.13)	
Gender								<0.0001
	Male	51.36 (50.29, 52.44)	54.60 (52.21, 57.10)	60.87 (57.22, 64.76)	67.87 (65.04, 70.82)	54.94 (54.30, 55.60)	43.47 (42.87, 44.07)	
	Female	48.64 (47.87, 49.42)	45.40 (43.11, 47.82)	39.13 (37.15, 41.21)	32.14 (27.31, 37.81)	45.06 (44.47, 45.65)	56.53 (56.04, 57.02)	
Marital Status								<0.0001
	Married	39.09 (38.07, 40.14)	35.36 (33.15, 37.72)	54.45 (50.96, 58.20)	8.02 (4.86, 13.27)	60.85 (60.40, 61.30)	55.59 (55.16, 56.02)	
	Divorced	19.36 (18.41, 20.37)	13.60 (11.05, 16.75)	10.58 (8.70, 12.86)	5.50 (3.03, 10.00)	12.82 (12.16, 13.52)	8.39 (7.93, 8.86)	
	Widowed	6.05 (5.47, 6.69)	3.80 (2.71, 5.32)	1.62 (1.02, 2.56)	1.59 (0.67, 3.79)	10.22 (9.77, 10.69)	6.30 (6.00, 6.62)	
	Separated	3.68 (2.95, 4.60)	4.31 (2.93, 6.34)	1.06 (0.37, 3.04)	0.00	1.47 (1.16, 1.86)	1.19 (0.98, 1.45)	
	Never Married	26.09 (24.90, 27.34)	35.74 (32.40, 39.41)	23.26 (20.17, 26.84)	76.75 (74.05, 79.53)	10.80 (9.97, 11.70)	25.55 (24.76, 26.35)	
	A member of an unmarried couple	5.72 (4.97, 6.59)	7.19 (5.37, 9.63)	9.03 (5.82, 13.99)	8.13 (5.41, 12.18)	3.84 (3.28, 4.50)	2.98 (2.67, 3.34)	
Employment								<0.0001
	Employed for wages	50.32 (49.30, 51.36)	51.66 (48.93, 54.55)	62.79 (59.98, 65.73)	59.15 (55.10, 63.50)	44.19 (43.42, 44.98)	53.44 (52.91, 53.96)	
	Self-employed	7.99 (7.14, 8.94)	7.40 (5.65, 9.69)	9.16 (5.60, 14.96)	3.82 (1.71, 8.51)	9.28 (8.68, 9.91)	8.63 (8.14, 9.16)	
	Out of work for 1 year or more	4.09 (3.43, 4.88)	5.63 (3.91, 8.12)	1.99 (1.20, 3.31)	1.28 (0.43, 3.81)	1.90 (1.63, 2.21)	1.77 (1.49, 2.09)	
	Out of work for less than 1 year	4.13 (3.50, 4.89)	3.71 (2.45, 5.62)	5.04 (2.63, 9.64)	2.57 (1.52, 4.34)	2.01 (1.49, 2.70)	2.28 (1.99, 2.61)	
	A homemaker	5.18 (4.43, 6.05)	4.88 (3.14, 7.58)	3.77 (2.26, 6.30)	1.56 (0.61, 4.03)	4.06 (3.63, 4.53)	5.63 (5.25, 6.03)	
	A student	1.56 (1.14, 2.12)	3.97 (2.39, 6.60)	3.08 (1.77, 5.37)	28.80 (23.90, 34.71)	1.17 (0.83, 1.64)	7.09 (6.52, 7.72)	
	Retired	12.23 (11.62, 12.87)	5.54 (4.62, 6.65)	6.78 (5.80, 7.93)	0.95 (0.34, 2.67)	30.32 (29.86, 30.79)	17.02 (16.66, 17.39)	
	Unable to work	14.51 (13.48, 15.61)	17.20 (14.78, 20.00)	7.40 (5.73, 9.57)	1.86 (0.63, 5.42)	7.08 (6.50, 7.70)	4.14 (3.78, 4.54)	
General Health								<0.0001
	Excellent	10.60 (9.66, 11.63)	10.79 (7.75, 15.02)	8.62 (7.22, 10.28)	29.92 (24.42, 36.67)	14.86 (14.17, 15.59)	21.45 (20.83, 22.10)	
	Very good	26.86 (25.76, 28.01)	22.72 (20.18, 25.58)	35.52 (31.39, 40.19)	35.70 (30.87, 41.28)	32.93 (32.20, 33.68)	37.30 (36.74, 37.88)	
	Good	34.55 (33.59, 35.53)	38.32 (35.64, 41.20)	32.81 (29.28, 36.77)	25.47 (21.14, 30.67)	32.44 (31.71, 33.18)	29.25 (28.64, 29.87)	
	Fair	19.30 (18.13, 20.53)	16.28 (14.19, 18.67)	16.25 (12.72, 20.75)	7.96 (5.93, 10.68)	13.93 (13.29, 14.60)	9.18 (8.72, 9.66)	
	Poor	8.70 (7.85, 9.64)	11.90 (9.77, 14.48)	6.81 (4.76, 9.74)	0.95 (0.40, 2.25)	5.84 (5.32, 6.41)	2.82 (2.48, 3.19)	
Mental Health	Average days mental health not good	6.09	8.06	5.16	5.45	2.76	2.49	<0.0001
Emotional Support								<0.0001
	Low	30.45 (29.18, 31.77)	32.91 (29.92, 36.21)	22.15 (18.40, 26.66)	19.80 (15.88, 24.72)	19.17 (18.36, 20.01)	17.40 (16.75, 18.06)	
	High	69.55 (69.04, 70.06)	67.09 (65.58, 68.63)	77.85 (76.42, 79.33)	80.20 (77.96, 82.50)	80.83 (80.59, 81.08)	82.60 (82.37, 82.84)	
Life Satisfaction								<0.0001
	Low	12.18 (11.04, 13.44)	15.88 (13.37, 18.86)	7.59 (5.34, 10.79)	8.05 (5.13, 12.63)	4.43 (3.99, 4.93)	3.40 (3.08, 3.75)	
	High	87.82 (87.62, 88.03)	84.12 (83.22, 85.03)	92.41 (91.74, 93.09)	91.95 (90.99, 92.92)	95.57 (95.49, 95.64)	96.60 (96.55, 96.65)	

Note: CI denotes confidence interval.

**Table 2 ijerph-19-10722-t002:** Association of life satisfaction with smoking/vaping status.

Low Life Satisfaction	Adjusted Odds Ratio (95% Confidence Interval)			
	All	Age: 18–34 (*n* = 7459)	Age: 35–64 (*n* = 24,141)	Age: 65+ (*n* = 15,563)
Dual users vs. Never users	**1.770 (1.135, 2.760)**	**2.719 (1.254, 5.896)**	1.280 (0.763, 2.146)	1.882 (0.682, 5.190)
Ex-smokers vs. Never users	1.025 (0.806, 1.303)	1.516 (0.806, 2.852)	0.915 (0.674, 1.244)	0.795 (0.535, 1.183)
Exclusive smokers vs. Never users	**1.452 (1.121, 1.880)**	**2.457 (1.439, 4.193)**	1.127 (0.829, 1.532)	1.170 (0.747, 1.832)
Current vapers who were ex-smokers vs. Never users	1.335 (0.708, 2.517)	1.334 (0.387, 4.594)	1.474 (0.679, 3.201)	**0.104 (0.014, 0.786)**
Exclusive vapers vs. Never users	1.230 (0.440, 3.440)	1.599 (0.544, 4.701)	0.190 (0.016, 2.206)	**0.013 (0.001, 0.161)**
Exclusive smokers vs. Ex-smokers	**1.416 (1.095, 1.831)**	1.621 (0.788, 3.332)	1.231 (0.908, 1.670)	1.471 (0.947, 2.285)

Note: *p*-values below 0.05 are in bold.

**Table 3 ijerph-19-10722-t003:** Association of social/emotional support with smoking/vaping status.

Low social/Emotional Support	Adjusted Odds Ratio (95% Confidence Interval)			
	All	Age: 18–34 (*n* = 7459)	Age: 35–64 (*n* = 24,141)	Age: 65+ (*n* = 15,563)
Dual users vs. Never users	1.075 (0.768, 1.505)	0.923 (0.498, 1.710)	1.169 (0.801, 1.708)	1.061 (0.563, 2.000)
Ex-smokers vs. Never users	0.933 (0.835, 1.042)	0.806 (0.557, 1.165)	0.944 (0.809, 1.102)	0.942 (0.801, 1.108)
Exclusive smokers vs. Never users	**1.193 (1.030, 1.381)**	1.243 (0.909, 1.698)	1.100 (0.906, 1.335)	**1.385 (1.061, 1.808)**
Current vapers who were ex-smokers vs. Never users	1.148 (0.713, 1.847)	0.939 (0.437, 2.017)	1.354 (0.697, 2.629)	0.896 (0.306, 2.626)
Exclusive vapers vs. Never users	0.762 (0.412, 1.407)	0.757 (0.390, 1.466)	0.708 (0.209, 2.397)	**0.014 (0.003, 0.071)**
Exclusive smokers vs. Ex-smokers	**1.279 (1.097, 1.492)**	**1.542 (1.003, 2.372)**	1.165 (0.949, 1.430)	**1.470 (1.125, 1.920)**

Note: *p*-values below 0.05 are in bold.

## Data Availability

The 2016 and 2017 BRFSS data are publicly available from the CDC website and can be downloaded from https://www.cdc.gov/brfss/annual_data/annual_data.htm (accessed on 16 February 2021).

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
