# Peer review of "Cross-Sectional Associations of Self-Reported Social/Emotional Support and Life Satisfaction with Smoking and Vaping Status in Adults"

_ijerph, 2022, doi:10.3390/ijerph191710722_

Round 1

Reviewer 1 Report

This paper using BRFSS data from 2016-2017 to examine the association between life satisfaction and social/emotional support within smoking or vaping status. However there does not appear to be strong rationale for including vaping in this analysis, and the discussion section should comment more on these lack of findings.

Aside from this research already being 5 years old, there are some places where the manuscript can be improved:

The introduction does a good job of outlining the research associated with social support and smoking cessation, but no evidence is provided to suggest that it would be associated with current smoking or vaping.  The rationale for including this analysis is weak.

Lines 78-82, run-on sentence, needs revision.

Sec. 2.2, need to address how other tobacco product use (cigar, smokeless, etc.) is accounted for or ignored in your 6 use categories. If ignored this limitation needs to be addressed in the discussion section.

Sec. 2.4, why those age categories. Most literature considers young adults 18-24. The range of possible  life events happening in the 18-34 category is likely to impact the life satisfaction results. 

Sec. 2.5, explain how weights were selected used if 2 years (2016 and 2017) of data were combined for the analysis?

Table 1, the order of the columns for the 6 categories is confusing. Try re-arranging based on continum of risk (exclusive smokers, dual users, vapers who are ex-smokers, exclusive vapers, ex-smokers, never users). Also you are missing important tobacco use information in this table- cigarettes per day, number of days used in the part moth, amount of e-liquid used per day, cigarette pack years, etc.

Discussion section- 

Line 207-208- too repetative, just repeats the opening sentence for that paragraph. 

Line 211-212, that statement "is consistent with our findings" is not supported by your results. your results did not look at adolescents, and was only cross-sectional not longitudinal.

Line 218-220, your study could have looked at the longitudinal change in life satisfaction when smokers quit smoking. Why was this not explored? Not clear what these cross-sectional analyses add to the literature. 

Paragraph starting line 225- Again focus is on relationship between low social support and cessation which this paper did not address. If this is of interest then additional analyses looking at this aim could be run. Although this is already well-established in the literature you cited.

Section 5 conclusions- Again your paper is not suggesting a relationship between the initiation of smoking or smoking cessation (line 261), but the relationship with current smoking status. You could possibly say your outcomes may be associated with smoking maintainence, and could be sources for intervention. But the continued extrapolation to smoking cessation seems unwarented based on the current cross-sectional analysis.

Author Response

Overview of responses to reviewers’ comments

Manuscript ID: ijerph-183820

We really appreciate the critical comments and suggestions from the reviewer. We have revised our manuscript based on the reviewer’s comments. The details are listed below point by point.

Reviewer 1

This paper using BRFSS data from 2016-2017 to examine the association between life satisfaction and social/emotional support within smoking or vaping status. However there does not appear to be strong rationale for including vaping in this analysis, and the discussion section should comment more on these lack of findings.

Response: While previous studies have been performed to examine the association of life satisfaction and social/emotional support with smoking, little is known about their associations with vaping as we mentioned in the Introduction “However, whether life satisfaction and social/emotional support might be associated with vaping remains undetermined”. Unfortunately, we did not identify any significant association of vaping with social/emotional support or life satisfaction, which might partially due to the relatively small sample size. As suggested, in the Discussion section we have added “In our study, we did not observe a significant association of social/emotional support with current vaping status, which could partially due to the relatively small sample size. Considering current vaping epidemic, it is important to determine its potential association with social/emotional support in future studies with large sample size”.

Aside from this research already being 5 years old, there are some places where the manuscript can be improved:

The introduction does a good job of outlining the research associated with social support and smoking cessation, but no evidence is provided to suggest that it would be associated with current smoking or vaping.  The rationale for including this analysis is weak.

Response: Thanks for the comments! There is one previous study on the association of social support and current smoking, “There is one study showed that adult smokers in South Korea rural areas had high levels of social support”. Because there is little evidence on the association of social support with current smoking and vaping to date, our current analysis aims to fill in this knowledge gap. We hypothesized that less social/emotional support might be associated with current smoking or vaping, which is actually supported by our analysis results.

Lines 78-82, run-on sentence, needs revision.

Response: Revised as suggested. “Since the survey data in 2016 and 2017 contain similar interview questions related to smoking and vaping, especially life satisfaction and emotional support, we decided to combine 2016 (486,303 participants) and 2017 (450,016 participants) BRFSS data in our analysis. The 2016 and 2017 BRFSS data is publicly available from the CDC website, which can be downloaded from https://www.cdc.gov/brfss/annual_data/annual_data.htm

Sec. 2.2, need to address how other tobacco product use (cigar, smokeless, etc.) is accounted for or ignored in your 6 use categories. If ignored this limitation needs to be addressed in the discussion section.

Response: Thanks for the comment. Unfortunately, BRFSS data do not provide the information about other tobacco product use. We have added it into the limitation as “In addition, there is no information on other tobacco product use such as cigar or smokeless tobacco product use in the BRFSS data. Thus, our study only considered smoking and vaping status, which could bias our results”.

Sec. 2.4, why those age categories. Most literature considers young adults 18-24. The range of possible life events happening in the 18-34 category is likely to impact the life satisfaction results. 

Response: It has been shown that compared to adults within other age categories, adults in the 18-34 category have higher odds of e-cigarette use as well as different socioeconomic statuses, such as poverty and marital status, which might affect their life satisfaction. We have added this explanation into the Method section.

Sec. 2.5, explain how weights were selected used if 2 years (2016 and 2017) of data were combined for the analysis?

Response: The final weight is a half of each year’ weight from the 2016 and 2017 BRFSS data to ensure the weighted sample size is equivalent to the population size. We added this in the Method section as “The final weight was a half of weight from each year (2016 and 2017) to ensure the weighted sample size is equivalent to the population size [8]”.

Table 1, the order of the columns for the 6 categories is confusing. Try re-arranging based on continuum of risk (exclusive smokers, dual users, vapers who are ex-smokers, exclusive vapers, ex-smokers, never users). Also you are missing important tobacco use information in this table- cigarettes per day, number of days used in the part moth, amount of e-liquid used per day, cigarette pack years, etc.

Response: The table has been re-arranged as suggested. We agree that these covariates are important tobacco use information. However, they are not provided in the BRFSS survey data. Therefore, we added this as one of the limitations, “Third, some other important information about smoking and vaping (such as if ever attempted to quit smoking, if there are current smokers or vapers in their family/friends, the number of cigarettes per day, the number of days used in the past month, and how long they have been smoking or vaping) are not included in our models due to the unavailability of these data in the BRFSS survey, which might introduce some biases into our final results”.

Discussion section- 

Line 207-208- too repetitive, just repeats the opening sentence for that paragraph. 

Response: Deleted as suggested.

Line 211-212, that statement "is consistent with our findings" is not supported by your results. your results did not look at adolescents, and was only cross-sectional not longitudinal.

Response: Thanks for the comment! We have removed “is consistent with our findings” as suggested.

Line 218-220, your study could have looked at the longitudinal change in life satisfaction when smokers quit smoking. Why was this not explored? Not clear what these cross-sectional analyses add to the literature. 

Response: Thanks for this valuable suggestion! Unfortunately, BRFSS data are not longitudinal data. Therefore, in this study we could not measure the longitudinal change in life satisfaction when smokers quit smoking. To reflect this, we have acknowledged this limitation in the Discussion section as “We also can not determine the longitudinal change in life satisfaction when smokers quit smoking”. Although our study is a cross-sectional study, it does provide some preliminary but important findings, which will prompt future longitudinal study.

Paragraph starting line 225- Again focus is on relationship between low social support and cessation which this paper did not address. If this is of interest then additional analyses looking at this aim could be run. Although this is already well-established in the literature you cited.

Response: Although our study did not directly measure the impact of social support on smoking cessation, our results did show that social support is significantly associated with current smoking status. That is, current smokers had low social support. Therefore, more social support might help with smoking cessation.

Section 5 conclusions- Again your paper is not suggesting a relationship between the initiation of smoking or smoking cessation (line 261), but the relationship with current smoking status. You could possibly say your outcomes may be associated with smoking maintenance, and could be sources for intervention. But the continued extrapolation to smoking cessation seems unwarranted based on the current cross-sectional analysis.

Response: Thanks for the comments! We revised as suggested, “the potential role of life satisfaction and social/emotional support in current smoking status. Together, our study results provided some preliminary but valuable information on potential smoking maintenance, which could be sources for possible interventions for smoking or even vaping cessation”.

Reviewer 2 Report

The manuscript investigates the association of self-reported social/emotional support and life satisfaction with six smoking and vaping status for overall US adults or by age group. Based on the combined BRFSS data in 2016-2017 (47,163 adult participants), the authors applied weighted logistic regression models to measure the adjusted odds ratios of five tobacco users vs. non-use for low life satisfaction and low social/emotional support by age group. As a whole, I found this manuscript well written with a clear structure and results, and the findings are meaningful for further studies of smoking and vaping disparity. However, I do have some comments and suggestions that I think need some attention.

Author Response

Overview of responses to reviewers’ comments

Manuscript ID: ijerph-183820

 We really appreciate the critical comments and suggestions from the  reviewer. We have revised our manuscript based on the reviewer’s comments. The details are listed below point by point.

The manuscript investigates the association of self-reported social/emotional support and life satisfaction with six smoking and vaping status for overall US adults or by age group. Based on the combined BRFSS data in 2016-2017 (47,163 adult participants), the authors applied weighted logistic regression models to measure the adjusted odds ratios of five tobacco users vs. non-use for low life satisfaction and low social/emotional support by age group. As a whole, I found this manuscript well written with a clear structure and results, and the findings are meaningful for further studies of smoking and vaping disparity. However, I do have some comments and suggestions that I think need some attention.

Response: Thanks for your encouraging comments on our manuscript!

Reviewer 3 Report

This study examined the cross-sectional association of self-reported social/emotional support and life satisfaction regarding smoking/vaping US adults for 47,163 adult participants who self-reported in the 2016 and 2017 BRFSS national survey data. The article is appropriately written and constructed. However, no consideration was given to whether in combining the 2016 and 2017 data the same respondents were being counted twice. As the only reason given for combining the two years was to increase the number of respondents, this is an important variable that the authors needed to address and did not. Furthermore, the articles referenced in most cases are out of date. The attitudes and practices of smokers and vapers change frequently. As a result, authors cannot rely on old references to support their claims—this is especially so if the authors are purporting to present cutting-edge research. Lastly, the references are not presented in MDPI style. They will need to be redone and the DOI number added to each article reference. As a final note, neither supplementary file was able to be downloaded.

Line by line suggested edits

29 Citation 1 is to a paper that cites this statistic to the US Department of Health and Human Services results from 2014:

US Department of Health and Human Services. The health consequences of smoking—50 years of progress: a report of the Surgeon General. Atlanta, GA: US Department of Health and Human Services, CDC; 2014. https://www.ncbi.nlm.nih.gov/books/NBK179276/pdf/Bookshelf_NBK179276.pdf 

More recent papers that have been published in peer-reviewed journals no longer cite smoking as the leading cause of preventable diseases and death. A search of Google Scholar for 2022 of “leading cause of death in adults” demonstrates that smoking is no longer mentioned as the leading cause of death:  https://scholar.google.ca/scholar?hl=en&as_sdt=0%2C5&as_ylo=2022&q=leading+cause+of+death+in+adults&btnG=

Please take note of this change in the leading cause of death in your article.

30 Again, citation 2 is to information published in 2014. It cannot be used to state what is the case in 2022.

40 References 9-11 are out of date. There are a number of new papers in peer reviewed journals that have been published on this matter in 2022. Please use more recent references.

41 Reference 12 is out of date. It is from 2002. A more recent reference is:

Cross, A. J.; Anthenelli, R.; Li, X. Metabotropic glutamate receptors 2 and 3 as targets for treating nicotine addiction. Bio. Psychi. 201883, 947-954. https://doi.org/10.1016/j.biopsych.2017.11.021

43 Change “smoking or vaping has” to smoking or vaping have”.

44 References 13 and 14 are from the previous century. Please don’t reference them. As well, there are more recent references available than the ones that are cited in 15 and 16 as seen from this Google Scholar search: https://scholar.google.ca/scholar?as_ylo=2018&q=smoking+or+vaping+associated+with+psychiatric+symptoms&hl=en&as_sdt=0,5

46 References 17-19 are out of date. Please cite more recent articles, such as : Ruggeri, K.; Garcia-Garzon, E.; Maguire, Á.; Matz, S.; Huppert, F.A. Well-being is more than happiness and life satisfaction: a multidimensional analysis of 21 countries. Health Quality Life Outcomes202018(1), 1-16. https://doi.org/10.1186/s12955-020-01423-y

and

Margolis, S.; Schwitzgebel, E.; Ozer, D.J.; Lyubomirsky, S. A new measure of life satisfaction: The Riverside Life Satisfaction Scale. J. Person. Assess. 2019101, 621-630. https://doi.org/10.1080/00223891.2018.1464457

48 Of references 20-25, only reference 20 is recent enough to be citable. This search of Google Scholar provides more recent references: https://scholar.google.ca/scholar?hl=en&as_sdt=0%2C5&as_ylo=2018&q=low+life+satisfaction+is+highly+associated+with+smoking%2C+marijuana+use%2C+alcohol+in+the+USA&btnG=

Note that one of the references (The Relationship between Life Satisfaction and Risk Behaviors: A Cross-Cultural Analysis of Youth) found the following, “On the other hand, there is no connection between life satisfaction and tobacco or marijuana use.”.

51 Reference 26 is from 1997. It cannot be cited to support data from 2018.

51-52 Need a reference for this claim,” Once addicted to nicotine, it is hard to stop due to the negative emotional responses involved in the withdrawal effect of nicotine”.

54 Reference 28 is out of date. Use a reference such as, Uchino, B. N.; Bowen, K.; Kent de Grey, R.; Mikel, J.; Fisher, E.B. (2018). Social support and physical health: Models, mechanisms, and opportunities. In Principles and concepts of behavioral medicine; Springer: New York, NY. 2018; pp. 341-372. Available at: https://link.springer.com/chapter/10.1007/978-0-387-93826-4_12

55-56 Change “cessation. While it remains controversial about the exact role of social and emotional support” to “cessation, although their exact role remains controversial”.

57 Change “tion[29” to “tion [29”.

All of references 29-31 are out of date—one is even from 1981. Articles must provide current information. This search from Google Scholar provides more recent references: https://scholar.google.ca/scholar?hl=en&as_sdt=0%2C5&as_ylo=2018&q=Social+and+emotional+support+in+smoking+cessation+is+controversial&btnG

58 Again, all of references 32-35 are out of date and cannot be used. Use this Google Scholar search to find more recent references: https://scholar.google.ca/scholar?hl=en&as_sdt=0%2C5&as_ylo=2018&q=positive+social+and+emotional+support+is+associated+with+successful+smoking+cessation%2C+relapse+prevention&btnG

60 References 33 and 34 are too old to support the claim.

65 -66 Need a reference for this claim: “vaping is more prevalent among young adults”.

80 In combining the 2016 and 2017 participants from the BRFSS data isn’t it likely that the same people in 2016 answered the questions in 2017? How can you be sure that the data from 2017 represent different responders than from 2016? If you can’t be sure, why is it acceptable to combine the responses to these two years?

159 Supporting information, Table S1 does not download. Nothing downloads from the supplementary files.

177 Supporting information, Table S2 does not download. Nothing downloads from the supplementary files.

212 As mentioned above, reference 20 is out of date.

230 All of these references are too old. Please find more recent references to support your claim, such as: Soulakova, J.N.; Tang, C.Y.; Leonardo, S.A.;  Taliaferro, L. A. Motivational benefits of social support and behavioural interventions for smoking cessation. J. Smoking Cess. 201813, 216-226. https://doi.org/10.1017/jsc.2017.26

232 References are out of date. This is an example of a current references: 

Mayne, S.L.; Auchincloss, A.H.; Moore, K.A.; Michael, Y.L.; Tabb, L.P.; Echeverria, S.E.; Roux, A.V.D. Cross-sectional and longitudinal associations of neighbourhood social environment and smoking behaviour: the multiethnic study of atherosclerosis. J Epidemiol Community Health 201771, 396-403. https://doi.org/10.1136/jech-2016-207990

232-234 Provide a recent reference for this claim.

234-235 This sentence is not clear and the reference is too old. Please eliminate this statement.

235-236 These references cited are too old and the idea of telephone quit-lines needs to be updated to online help. 

237 This reference is too old. You will have to find a reference that is US based and related to the time-period in which the data were collected, otherwise, you need to make this claim historically.

239 This reference is much to old. A recent one is: Whitton, S.W.; McLeish, A.C.; Godfrey, L.M.; James-Kangal, N.; Rhoades, G.K. Partner assisted smoking cessation treatment: a randomized clinical trial. Substance Use Misuse 202055, 1228-1236. https://doi.org/10.1080/10826084.2020.1731548

239-242 Need a reference for this claim. 

Author Response

Overview of responses to reviewers’ comments

Manuscript ID: ijerph-183820

We really appreciate the critical comments and suggestions from the reviewer. We have revised our manuscript based on the reviewer’s comments. The details are listed below point by point.

Reviewer 3

This study examined the cross-sectional association of self-reported social/emotional support and life satisfaction regarding smoking/vaping US adults for 47,163 adult participants who self-reported in the 2016 and 2017 BRFSS national survey data. The article is appropriately written and constructed. However, no consideration was given to whether in combining the 2016 and 2017 data the same respondents were being counted twice. As the only reason given for combining the two years was to increase the number of respondents, this is an important variable that the authors needed to address and did not.

Response: Thanks for the valuable comments. The BRFSS survey are anonymous, thus, we are not able to identify which participants answered both the 2016 and 2017 BRFSS survey questionnaire. We agree that there might be some overlap in participants between the 2016 and 2017 BRFSS survey, which could bring some bias to our analysis results. We have added this limitation into the Discussion section “There is a possibility that some participants might answered both the 2016 and 2017 BRFSS survey questionnaire. However, we are not able to identify those participants due to the anonymous property of the BRFSS survey, which might bring some bias to our analysis results.” In addition, the BRFSS survey was designed in a way such that data from different years could be combined to increase the sample size. Several studies have been using the combined BRFSS data from different years to study the association of e-cigarette use with respiratory diseases (such as COPD and Asthma) and other health conditions. We have added this into the revised Method section.

Furthermore, the articles referenced in most cases are out of date. The attitudes and practices of smokers and vapers change frequently. As a result, authors cannot rely on old references to support their claims—this is especially so if the authors are purporting to present cutting-edge research.

Response: As suggested, we have updated references with more recent articles.

Lastly, the references are not presented in MDPI style. They will need to be redone and the DOI number added to each article reference.

Response: Reformat the references according to the guideline provided by MDPI.

As a final note, neither supplementary file was able to be downloaded.

Response: Both supplemental tables have been uploaded.

Line by line suggested edits

29 Citation 1 is to a paper that cites this statistic to the US Department of Health and Human Services results from 2014:

US Department of Health and Human Services. The health consequences of smoking—50 years of progress: a report of the Surgeon General. Atlanta, GA: US Department of Health and Human Services, CDC; 2014. https://www.ncbi.nlm.nih.gov/books/NBK179276/pdf/Bookshelf_NBK179276.pdf 

More recent papers that have been published in peer-reviewed journals no longer cite smoking as the leading cause of preventable diseases and death. A search of Google Scholar for 2022 of “leading cause of death in adults” demonstrates that smoking is no longer mentioned as the leading cause of death:  https://scholar.google.ca/scholar?hl=en&as_sdt=0%2C5&as_ylo=2022&q=leading+cause+of+death+in+adults&btnG=

Please take note of this change in the leading cause of death in your article.

Response: Thanks for the update! We have deleted this sentence.

30 Again, citation 2 is to information published in 2014. It cannot be used to state what is the case in 2022.

Response: We have updated this sentence as “Tobacco smoking kills more than 8 million people each year worldwide [1]”.

40 References 9-11 are out of date. There are a number of new papers in peer reviewed journals that have been published on this matter in 2022. Please use more recent references.

Response: Updated as suggested.

41 Reference 12 is out of date. It is from 2002. A more recent reference is:

Cross, A. J.; Anthenelli, R.; Li, X. Metabotropic glutamate receptors 2 and 3 as targets for treating nicotine addiction. Bio. Psychi. 201883, 947-954. https://doi.org/10.1016/j.biopsych.2017.11.021

Response: Updated as suggested.

43 Change “smoking or vaping has” to smoking or vaping have”.

Response: Revised as suggested.

44 References 13 and 14 are from the previous century. Please don’t reference them. As well, there are more recent references available than the ones that are cited in 15 and 16 as seen from this Google Scholar search: https://scholar.google.ca/scholar?as_ylo=2018&q=smoking+or+vaping+associated+with+psychiatric+symptoms&hl=en&as_sdt=0,5

Response: Updated as suggested.

46 References 17-19 are out of date. Please cite more recent articles, such as : Ruggeri, K.; Garcia-Garzon, E.; Maguire, Á.; Matz, S.; Huppert, F.A. Well-being is more than happiness and life satisfaction: a multidimensional analysis of 21 countries. Health Quality Life Outcomes202018(1), 1-16. https://doi.org/10.1186/s12955-020-01423-y

and

Margolis, S.; Schwitzgebel, E.; Ozer, D.J.; Lyubomirsky, S. A new measure of life satisfaction: The Riverside Life Satisfaction Scale. J. Person. Assess. 2019101, 621-630. https://doi.org/10.1080/00223891.2018.1464457

Response: Updated as suggested.

48 Of references 20-25, only reference 20 is recent enough to be citable. This search of Google Scholar provides more recent references: https://scholar.google.ca/scholar?hl=en&as_sdt=0%2C5&as_ylo=2018&q=low+life+satisfaction+is+highly+associated+with+smoking%2C+marijuana+use%2C+alcohol+in+the+USA&btnG=

Note that one of the references (The Relationship between Life Satisfaction and Risk Behaviors: A Cross-Cultural Analysis of Youth) found the following, “On the other hand, there is no connection between life satisfaction and tobacco or marijuana use.”.

Response: Updated as suggested.

51 Reference 26 is from 1997. It cannot be cited to support data from 2018.

Response: Reference 26 has been removed.

51-52 Need a reference for this claim,” Once addicted to nicotine, it is hard to stop due to the negative emotional responses involved in the withdrawal effect of nicotine”.

Response: Added as suggested!

54 Reference 28 is out of date. Use a reference such as, Uchino, B. N.; Bowen, K.; Kent de Grey, R.; Mikel, J.; Fisher, E.B. (2018). Social support and physical health: Models, mechanisms, and opportunities. In Principles and concepts of behavioral medicine; Springer: New York, NY. 2018; pp. 341-372. Available at: https://link.springer.com/chapter/10.1007/978-0-387-93826-4_12

Response: Updated as suggested.

55-56 Change “cessation. While it remains controversial about the exact role of social and emotional support” to “cessation, although their exact role remains controversial”.

Response: Revised as suggested.

57 Change “tion[29” to “tion [29”.

Response: Done.

All of references 29-31 are out of date—one is even from 1981. Articles must provide current information. This search from Google Scholar provides more recent references: https://scholar.google.ca/scholar?hl=en&as_sdt=0%2C5&as_ylo=2018&q=Social+and+emotional+support+in+smoking+cessation+is+controversial&btnG

Response: Updated as suggested.

58 Again, all of references 32-35 are out of date and cannot be used. Use this Google Scholar search to find more recent references: https://scholar.google.ca/scholar?hl=en&as_sdt=0%2C5&as_ylo=2018&q=positive+social+and+emotional+support+is+associated+with+successful+smoking+cessation%2C+relapse+prevention&btnG

Response: Updated as suggested.

60 References 33 and 34 are too old to support the claim.

Response: This claim is deleted.

65 -66 Need a reference for this claim: “vaping is more prevalent among young adults”.

Response: Added.

80 In combining the 2016 and 2017 participants from the BRFSS data isn’t it likely that the same people in 2016 answered the questions in 2017? How can you be sure that the data from 2017 represent different responders than from 2016? If you can’t be sure, why is it acceptable to combine the responses to these two years?

Response: Please refer to our above responses to the first comment.

159 Supporting information, Table S1 does not download. Nothing downloads from the supplementary files.

177 Supporting information, Table S2 does not download. Nothing downloads from the supplementary files.

Response: We have uploaded both supporting tables.

212 As mentioned above, reference 20 is out of date.

Response: Updated as suggested.

230 All of these references are too old. Please find more recent references to support your claim, such as: Soulakova, J.N.; Tang, C.Y.; Leonardo, S.A.;  Taliaferro, L. A. Motivational benefits of social support and behavioural interventions for smoking cessation. J. Smoking Cess. 201813, 216-226. https://doi.org/10.1017/jsc.2017.26

Response: Updated as suggested.

232 References are out of date. This is an example of a current references: 

Mayne, S.L.; Auchincloss, A.H.; Moore, K.A.; Michael, Y.L.; Tabb, L.P.; Echeverria, S.E.; Roux, A.V.D. Cross-sectional and longitudinal associations of neighbourhood social environment and smoking behaviour: the multiethnic study of atherosclerosis. J Epidemiol Community Health 201771, 396-403. https://doi.org/10.1136/jech-2016-207990

Response: Updated as suggested.

232-234 Provide a recent reference for this claim.

Response: Added.

234-235 This sentence is not clear and the reference is too old. Please eliminate this statement.

Response: Deleted as suggested.

235-236 These references cited are too old and the idea of telephone quit-lines needs to be updated to online help. 

Response: Updated as suggested, “It has been shown that the online help becomes popular for providing social and emotional support for smokers”.

237 This reference is too old. You will have to find a reference that is US based and related to the time-period in which the data were collected, otherwise, you need to make this claim historically.

Response: Revised as suggested “While it was reported that most smokers have attempted to quit smoking on their own…”.

239 This reference is much to old. A recent one is: Whitton, S.W.; McLeish, A.C.; Godfrey, L.M.; James-Kangal, N.; Rhoades, G.K. Partner assisted smoking cessation treatment: a randomized clinical trial. Substance Use Misuse 202055, 1228-1236. https://doi.org/10.1080/10826084.2020.1731548

Response: Updated as suggested.

239-242 Need a reference for this claim. 

Response: Added as suggested.

Round 2

Reviewer 1 Report

The authors addressed the major concerns outline in the first review. There still could be more discussion around the large age range used for young adults (ex, an 18 year old may experience very different life satisfaction/ social support than a 34 year old). Also the author's note their null findings between vaping and their outcomes of interest may be due to low sample size, but it seems equally plausible that there is just not a similar association that is seen in smokers. 

Author Response

Comments and Suggestions for Authors

The authors addressed the major concerns outline in the first review. There still could be more discussion around the large age range used for young adults (ex, an 18 year old may experience very different life satisfaction/ social support than a 34 year old). Also the author's note their null findings between vaping and their outcomes of interest may be due to low sample size, but it seems equally plausible that there is just not a similar association that is seen in smokers. 

Response: Thanks for the valuable comments on our revision. We agree that the large age range used in this study could introduce some biases. Therefore, we have added some discussion in the revised manuscript as “Finally, in this study we have grouped survey participants into three large age groups. However, the level of life satisfaction and social/emotional support depends on the age [60] [61]. Even within the same age group (such as age group 18-34) participants with different ages might have different levels of life satisfaction and social/emotional support [60]. Therefore, our age grouping method might introduce some biases”.

In addition, as suggested, for no significant association between vaping and the outcomes of interest, we have added in our discussion “A lack of association of vaping with life satisfaction could be due to either relatively small sample size or the possibility that there is no association between vaping and life satisfaction” and “Another possible explanation is that unlike smoking, vaping is not associated with social/emotional support”.

Reviewer 3 Report

This is the second review of a study examining the association of self-reported life satisfaction and social/emotional support with smoking and vaping status using the US The Behavioral Risk Factor Surveillance System. The study results are intended to inform the development of future interventions to promote both smoking and vaping prevention and/or cessation among US adults. The changes the authors have made have greatly improved the paper. However, although the authors claimed to have uploaded the supplementary files, when I try to download them, there is nothing that downloads. As such, I still have not seen the supplementary files. 

Now that the major changes requested have been made by the authors, there remain a few more changes to be incorporated. These are mentioned in the following line by line suggested edits.

Line by line suggested edits

11-12 Delete “It is estimated that about 18.1% are smokers and 6% are electronic cigarette users (vapers) in the US.”

34-36 Need a reference for the claim that there has been a decline in smoking prevalence in recent years and another reference for the claim that vaping has become increasingly popular especially among youths.

46 Change “smoking or vaping have” to “smoking and vaping have both”.

57 Change “receives empathetic” to “receives as a result of empathetic”.

59 Change “is usually regarded as an essential factor” to “are usually regarded as essential factors

61 Change “showed” to “have shown”.

73 Change “will inform” to “are intended to inform”.

74 Change “smoking or vaping prevention and or” to “both smoking and vaping prevention and/or”.

79 Change “ conducted” to “, conducted”.

85 Change “combined, which have been used to” to ”combined to”.

87 Please explain why the data from 2016 and 2017 were used instead of data from more recent years.  If this this the most recent data available, please say so.

91 Change “, which can be downloaded from” to “:”

152 I’m not sure what you mean by “other”. Can this word be left out? Or, do you mean the other five smoking/vaping categories, as mentioned later, in line 172?

Table 1 Please widen the size of the first column so that “Employment” and “Satisfaction”  appear on one row.

233-234 “low life satisfaction might motivate individuals (especially young adults) to engage in health risk-related behaviors, such as tobacco use, alcohol, and drug abuse.” You need a reference for this claim.

235 “On the other hand, these health risk behaviors might lead to low life satisfaction.” You need a reference for this claim.

256 Change “becomes” to “is”.

269 “vaping epidemic”—Need a reference for this claim.

288 Change “might answered” to “might have answered”.

297 Delete “possible”.

316 Delete “Acknowledgment: None”.

References

The following references are too old and require up to date references: 22, 23 and 26.

Author Response

Comments and Suggestions for Authors

This is the second review of a study examining the association of self-reported life satisfaction and social/emotional support with smoking and vaping status using the US The Behavioral Risk Factor Surveillance System. The study results are intended to inform the development of future interventions to promote both smoking and vaping prevention and/or cessation among US adults. The changes the authors have made have greatly improved the paper. However, although the authors claimed to have uploaded the supplementary files, when I try to download them, there is nothing that downloads. As such, I still have not seen the supplementary files. 

Response: Not sure what went wrong. We have uploaded the supplemental files.

Now that the major changes requested have been made by the authors, there remain a few more changes to be incorporated. These are mentioned in the following line by line suggested edits.

Line by line suggested edits

11-12 Delete “It is estimated that about 18.1% are smokers and 6% are electronic cigarette users (vapers) in the US.”

Response: Done as suggested.

34-36 Need a reference for the claim that there has been a decline in smoking prevalence in recent years and another reference for the claim that vaping has become increasingly popular especially among youths.

Response: As suggested, we have added one reference on the decline in smoking prevalence, and another one on the increase in vaping prevalence.

46 Change “smoking or vaping have” to “smoking and vaping have both”.

Response: Done as suggested.

57 Change “receives empathetic” to “receives as a result of empathetic”.

Response: Done as suggested.

59 Change “is usually regarded as an essential factor” to “are usually regarded as essential factors”

Response: Done as suggested.

61 Change “showed” to “have shown”.

Response: Done as suggested.

73 Change “will inform” to “are intended to inform”.

Response: Done as suggested.

74 Change “smoking or vaping prevention and or” to “both smoking and vaping prevention and/or”.

Response: Done as suggested.

79 Change “conducted” to “, conducted”.

Response: Done as suggested.

85 Change “combined, which have been used to” to ”combined to”.

Response: Done as suggested.

87 Please explain why the data from 2016 and 2017 were used instead of data from more recent years.  If this this the most recent data available, please say so.

Response: The survey data in 2016 and 2017 are the most recent BRFSS survey data that contain main survey question about life satisfaction and social/emotional support. We have added “Since the survey data in 2016 and 2017 contain the most recent and similar interview questions related to smoking and vaping, especially life satisfaction and emotional support, we decided to combine 2016 (486,303 participants) and 2017 (450,016 participants) BRFSS data in our analysis”.

91 Change “, which can be downloaded from” to “:”

Response: Done as suggested.

152 I’m not sure what you mean by “other”. Can this word be left out? Or, do you mean the other five smoking/vaping categories, as mentioned later, in line 172?

Response: Thanks for the comments. We have changed it to “the other five smoking/vaping categories".

Table 1 Please widen the size of the first column so that “Employment” and “Satisfaction” appear on one row.

Response: Done as suggested.

233-234 “low life satisfaction might motivate individuals (especially young adults) to engage in health risk-related behaviors, such as tobacco use, alcohol, and drug abuse.” You need a reference for this claim.

Response: Done as suggested.

235 “On the other hand, these health risk behaviors might lead to low life satisfaction.” You need a reference for this claim.

Response: Done as suggested.

256 Change “becomes” to “is”.

Response: Done as suggested.

269 “vaping epidemic”—Need a reference for this claim.

Response: Done as suggested.

288 Change “might answered” to “might have answered”.

Response: Revised as suggested.

297 Delete “possible”.

Response: Done as suggested.

316 Delete “Acknowledgment: None”.

Response: Done as suggested.

References

The following references are too old and require up to date references: 22, 23 and 26.

Response: Updated as suggested.